# Vaccine or Garlic–Is It a Choice? Awareness of Medical Personnel on Prevention of Influenza Infections

**DOI:** 10.3390/vaccines11010066

**Published:** 2022-12-28

**Authors:** Tomasz Sobierajski, Dominika Rykowska, Monika Wanke-Rytt, Ernest Kuchar

**Affiliations:** 1Faculty of Applied Social Sciences and Resocialization, University of Warsaw, 26/28 Krakowskie Przedmieście Str., 00-927 Warsaw, Poland; 2Department of Pediatrics with Clinical Assessment Unit, Medical University of Warsaw, 63a Żwirki & Wigury Str., 02-091 Warsaw, Poland

**Keywords:** physicians, nurses, hospital, mask, vitamin C, pediatric, inosine, immunization

## Abstract

Background: Preventing the spread of the influenza virus is one of the primary health policy challenges of many countries worldwide. One of the more effective ways to prevent infection is influenza vaccination, and the people who enjoy the most public confidence in preventive health care are health workers (HWs). For this reason, it is crucial to study the attitudes of HWs toward influenza vaccination. Methods: The survey was conducted among 950 medical (physicians and nurses) and administrative staff in three academic hospitals. Respondents to the survey were selected on a random-target basis to represent hospital employees in the study best. The survey was conducted using the PAPI method between August and September 2020. Results: Respondents considered hand washing (52.8%) and avoiding contact with sick people (49.3%) the most effective ways to prevent influenza infection. Three in ten respondents considered wearing a protective mask (30.1%) and getting vaccinated against influenza (29.9%) is fully effective in preventing influenza. Influenza vaccination as effective in preventing influenza virus infection was chosen more often by those who worked in a pediatric hospital. Nurses were twice less likely than physicians to declare that influenza vaccination prevents infection (42.4% for nurses vs. 84.0% for physicians). At the same time, 20.4% of nurses believed that eating garlic effectively prevented influenza infection, and 28.1% declared daily vitamin C helpful. Conclusions: The study pointed to significant educational gaps regarding the role and effectiveness of influenza vaccination in the process of influenza virus infection and indicated a firm belief in medical myths, especially in the nursing community, related to protection against influenza virus infection.

## 1. Introductions

Seasonal influenza is a severe health problem worldwide. An estimated one billion people contract influenza yearly, including most children and the elderly. Among all cases, several million are severe cases with post-influenza complications, and nearly half a million die annually from respiratory illnesses resulting from influenza [1]. Worldwide, 10% of all hospitalizations from respiratory infections in children are caused by the influenza virus [2,3]. Among adults, influenza is associated with over five million hospitalizations per year globally [4]. To decrease influenza’s social and economic burden, the WHO created the Global Influenza Strategy 2019–2030, where influenza prevention is one of the main goals [5]. According to the strategy, among other things, it is crucial to identify the need for rapid health prevention interventions against influenza with particular attention to knowledge gaps about the influenza virus and to increase public confidence in influenza vaccine uptake. 

Health workers (HWs) are some of the leading educators in health prevention, including knowledge of influenza virus and influenza vaccination. Public trust in physicians and nurses translates or can translate into health-promoting behavior by patients. Physicians and nurses should inform patients about ways to prevent contracting influenza during patient encounters. Ideally, influenza vaccination should be the most recommended method of influenza prevention. However, a recently published article showed a gap in general knowledge about the influenza virus in the Polish healthcare workers population [6]. For this reason, it seems so essential to know the level of HWs’ knowledge of ways to prevent influenza virus infection: both those that are confirmed by scientific research (vaccination, hand washing) and those whose effectiveness is not confirmed by scientific research but are still firmly rooted in the public consciousness (taking vitamin C, eating garlic). It is because the knowledge of HWs translates into their beliefs and, as a result, can translate into educational dispositions in preventive health care that are passed on to patients. 

To our knowledge, studies have yet to be conducted in Poland that have explored to such a degree and scaled the attitudes and knowledge of HWs toward ways to prevent influenza virus infection. 

Based on the available literature and the clinical experience of the article’s authors, we adopted the following research hypotheses for our study, which we verified during our research.

**H1.** *Physicians believe more in clinical methods (washing hands, wearing a mask covering mouth and nose, vaccination) to prevent influenza virus infection than nurses*.

**H2.** *Nurses and other (non-medical) hospital personnel show greater belief than physicians in the effectiveness of non-clinical ways to prevent influenza virus infection, such as eating garlic and taking vitamin C daily*. 

**H3.** *Influenza vaccination is perceived to prevent influenza virus infection more often among pediatric hospital staff than adult hospital staff and physicians than nurses*.

## 2. Materials and Methods

### 2.1. Study Design

Warsaw Medical University (WUM) is Poland’s largest public university training medical personnel. Three academic hospitals at WUM (one children’s hospital and two adult hospitals) make up the University Clinical Center of the Warsaw Medical University (UCK WUM). These three hospitals provide training for students of all WUM faculties. HWs employed at the UCK WUM are formal or informal teachers for WUM students and residents, i.e., young physicians and nursing staff. For this reason, investigating the level of knowledge of HWs from UCK WUM about ways to prevent influenza virus infection was crucial to the entire research process.

### 2.2. Sample Size and Characteristics

The study population consisted of all employees (medical and non-medical) of the three UCK WUM hospitals. At the time of the survey, 1643 HWs and 297 non-medical employees worked at the pediatric hospital, and 5443 HWs and 1038 non-medical employees worked at the adult hospitals. In total, the study population was 8421. A total of 1233 employees participated in the survey. Only those correctly completed questionnaires were approved for further analysis, and the final study sample consisted of 950 people. We calculated the sample size, assuming a margin of error of no more than 5% at the 95% confidence level.

The respondents were selected by a random-target method. Before the survey was implemented, three groups of respondents were identified: physicians, nurses, and other hospital employees. The survey was implemented in all hospital departments to increase the diversity of those participating in the study. One of the study’s authors implemented the survey and personally contacted and surveyed the respondents. The survey also considered the quotas of respondents. In implementing the survey, we ensured that the percentage of respondents in each of the three groups was close to the percentage of each group in the general survey population.

### 2.3. The Questionnaire

The questionnaire was divided into several parts. The first metric part included questions about gender, occupation, age, length of service, and type of hospital. The second part included five questions about, among other things, having been vaccinated against influenza in the past season, having contact with immunosuppressed patients, and planning to be vaccinated against influenza in the next season. In this part, all questions were dichotomous (answers: I do or I do not) or alternatively (answers: I do, I do not, I do not know). The third part included three questions. Two were about knowledge, and one was about attitudes toward influenza vaccination. In the first question in this part, we asked about seasonal influenza vaccination, and in the second one, we asked about the effectiveness of methods to prevent influenza virus infection. The question on attitudes was a design question relating to respondents’ motivations concerning influenza vaccination. Responses to each question in this questionnaire section were placed on a numerical, seven-point Likert scale. The extremes of scale for questions about knowledge meant: from 1-strongly disagree to 7-strongly agree and for the question about the effectiveness of methods to prevent influenza virus infection: from 1-fully ineffective to 7-fully effective. 

The survey questionnaire was evaluated using a pilot study to verify the tool’s effectiveness. Twenty-three people participated in the pilot study: seven physicians, eleven nurses, and five other hospital employees. Evaluation of the tool was a necessary methodological procedure, to indicate the usefulness and relevance of the questions used in the questionnaire. Both usefulness and relevance of questionnaire were fully confirmed during the evaluation. Furthermore, none of the people who participated in the questionnaire evaluation objected to the content of the questions and the phrases used in the questionnaire.

It took respondents about 15 min to complete the questionnaire.

### 2.4. Data Collection

The survey was conducted in September/October 2020. The survey was conducted using the PAPI (Paper and Pen Interview) technique. Before the survey began, respondents were informed that the survey was anonymous and confidential. The data were recorded on an ongoing basis in a database to which only the survey authors had access.

### 2.5. Statistical Analysis

All statistical analyses were carried out using IBM SPSS Statistics 28.0.1.1. Descriptive analysis was performed to describe the sample, and the results were presented as frequencies and percentages, and chi-square was used to compare frequencies. The scales in the questionnaire were validated using Cronbach’s alpha test, and normality was calculated using the Shapiro-Wilk test. 

### 2.6. Ethical Considerations

The Ethics Committee of the Medical University of Warsaw approved the study protocol, permission number AKBE/118/2020.

## 3. Results

### 3.1. Sociodemographic Characteristics

Most of the 950 respondents, 84.9% (*N* = 807) were women. Age groups between 18–60 were evenly represented, while those over 60 were the least numerous. One in three respondents was physician (32.2%). Over half of the people surveyed (55.6) worked in a children’s hospital. Four out of ten had work experience of more than 20 years (Table 1).

### 3.2. Acceptance of Methods to Prevent Influenza Infections

Respondents rated the effectiveness of the seven potential methods to prevent influenza virus infection differently. Respondents considered frequent hand washing the most effective way to prevent influenza infection; half of the respondents considered it fully effective (52.8%, *N* = 502). Half of the respondents also considered avoiding contact with sick people fully effective (49.3%, *N* = 468). Three in ten respondents considered wearing a protective mask that covers the mouth and nose to be fully effective in preventing influenza (30.1%, *N* = 286) and getting vaccinated against influenza (29.9%, *N* = 284). Respondents considered taking vitamin C daily to be the least effective, although 8.6% (*N* = 82) considered it fully effective. A quantity of 6.4% (*N* = 61) of respondents considered eating garlic fully effective, and 3.4% (*N* = 32) considered using inosine preparations fully effective (Table 2).

### 3.3. Method 1: Mask Covering Mouth and Nose

A protective mask covering the mouth and nose to prevent influenza virus infection was rated differently according to gender, age, and profession (Table 3). A protective mask covering the mouth and nose was rated as completely effective by 30.6% (*N* = 247) of women and 27.3% (*N* = 39) of men (Figure 1). The younger the respondent, the rating of the total effectiveness of this method decreased. While one in three respondents over the age of 50 (36.8%, *N* = 25 for those over 60 and 36.2%, *N* = 80, for those between 51 and 60) found the mask completely effective, one in five respondents aged 18–30 found it completely effective (22.2%, *N* = 49) (*p* = 0.015). At the same time, one in nine respondents over 60 (11.8%, *N* = 8) considered wearing a mask to be completely ineffective in preventing influenza virus infection (Figure 2). Wearing a mask was considered completely effective by a similar percentage of respondents in both types of hospitals (30.6%, *N* = 129 in adult hospitals and 29.7%, *N* = 157 in a children’s hospital) (Figure 3). Wearing a mask was considered completely effective by one in three respondents, regardless of the type of profession (27.8%, *N* = 85 for physicians; 31.1%, *N* = 142 for nurses; 31.4%, *N* = 59 for other professions) (*p* < 0.001). Wearing a mask was considered completely ineffective by 2.3% (*N* = 7) of physicians and 8.3% (*N* = 38) of nurses (Figure 4) (Figures 36 and 37). The higher the seniority, the more effective wearing a mask was considered (Figure 5). It was considered completely effective by 36.5% (*N* = 142) of respondents with more than 20 years of seniority vs. 24.9% (*N* = 61) of respondents with up to 5 years of seniority (*p* = 0.003).

### 3.4. Method 2: Hand Washing

Hand washing to prevent influenza virus infection was rated differently according to seniority, age, and profession (Table 4). Hand washing was considered completely effective by 53.5% (*N* = 432) of women and 49.0% (*N* = 70) of men (Figure 6). Handwashing as completely effective was most often recognized by respondents aged 40–60 (59.0%, *N* = 138 for those 41–50 and 57.9%, *N* = 128 for those 51–60) and least often by respondents aged 18–30 (44.8%, *N* = 99) (*p* < 0.001) (Figure 7). Hand washing was considered completely effective by 55.2% (*N* = 233) of those working in adult hospitals and 50.9% (*N* = 269) of those working in a pediatric hospital (*p* < 0.001) (Figure 8). Those working in other professions were more likely than physicians and nurses to believe that hand washing completely prevents getting influenza (61.2%, *N* = 115 for others; 53.3%, *N* = 243 for nurses; 47.1%, *N* = 144 for physicians) (*p* < 0.001) (Figure 9) (Figures 36 and 37). With seniority, the belief that handwashing prevents influenza virus infection increases—it is considered completely effective most often by those working more than 20 years (59.6%, *N* = 232) and least often by those working less than 5 years (46.1%, *N* = 113) (*p* < 0.001) (Figure 10).

### 3.5. Method 3: Influenza Vaccination

Influenza vaccination to prevent influenza virus infection was rated differently according to seniority, type of hospital, age, and profession (Table 5). Influenza vaccination as completely effective was considered significantly more often by men than women (28.0%, *N* = 226 for women vs. 40.6%, *N* = 58 for men) (*p* = 0.002) (Figure 11). Vaccination as completely effective was the least frequently considered by those aged 41–50 (21.4%, *N* = 50). In the remaining age groups, an average of one in three respondents considered influenza vaccination to be completely effective. Younger respondents under 40 were twice as likely to consider influenza vaccination completely ineffective as respondents over 40 (5%, *N* = 11 for 18–30-year-olds; 5.8%, *N* = 12 for 31–40-year-olds; 11.5%, *N* = 27 for 41–50-year-olds; 10.0%, *N* = 22 for 51–60-year-olds; 13.2%, *N* = 9 for those over 60) (*p* < 0.001) (Figure 12). Influenza vaccination as a means of preventing influenza illness was considered completely effective significantly more often by those working in a pediatric hospital (34.7%, *N* = 183) than in an adult hospital (23.9%, *N* = 101). At the same time, one in eight people from a pediatric hospital (12.1%, *N* = 51) and 5.7% (*N* = 30) respondents from an adult hospitals (*p* < 0.001) consider influenza vaccination to prevent getting influenza (Figure 13). One in two physicians (49.3%, *N* = 151), one in five nurses (20.8%, *N* = 95), and one in five others (20.2%, *N* = 38) consider influenza vaccination as completely effective in preventing influenza illness. One in eight nurses (12.9%, *N* = 59), one in ten others (10.1%, *N* = 19) and 1.0% (*N* = 3) of physicians (*p* < 0.001) consider vaccination to be completely ineffective (Figure 14) (Figures 36 and 37). Influenza vaccination is considered completely effective by one in five people who did not specify the length of service (19.2%, *N* = 19). Elsewhere in this category, an average of one in three people considers it completely effective. Influenza vaccination is considered completely ineffective most often by those with seniority of more than 20 years (10.5%, *N* = 41) and those who did not specify their seniority in their profession (11.1%, *N* = 11) (*p* < 0.001) (Figure 15).

The results presented above regarding clinical approaches to preventing influenza virus infection and attitudes toward the effectiveness of influenza vaccination as a form of preventing influenza virus infection positively verify hypotheses H1 and H3.

**H1.** *Physicians show greater belief in clinical methods (hand washing, wearing a mask covering the mouth and nose, vaccination) of preventing influenza virus infection than nurses*.

**H3.** *Influenza vaccination is perceived to prevent influenza virus infection more often among pediatric hospital staff than adult hospital staff and physicians than nurses*.

### 3.6. Method 4: Avoiding Contact with Sick People

Avoiding contact with sick people to prevent influenza virus infection was rated differently by gender, age, and length of service (Table 6). Avoiding contact with sick people as completely effective was considered slightly more often by women than men (50.2%, *N* = 405 for women vs. 44.1%, *N* = 63 for men) (Figure 16). Avoiding contact with sick people as completely effective is considered equally by people of all ages (one in two people on average). However, it is considered completely ineffective by one in eight people over the age of 60 (13.2%, *N* = 9) (*p* = 0.014) (Figure 17). Avoiding contact with sick people as a means of preventing influenza virus infection is slightly more likely to be considered completely effective by those working in a pediatric hospital (52.5%, *N* = 277) than those working in an adult hospital (45.3%, *N* = 191) (*p* < 0.001) (Figure 18). Avoiding contact with sick people is considered completely effective by an average of one in two people regardless of the profession (Figure 19) (Figures 36 and 37) and an average of one in two people regardless of seniority (*p* < 0.001) (Figure 20).

### 3.7. Method 5: Eating Garlic

Eating garlic to prevent influenza virus infection was rated differently according to seniority, age, and profession (Table 7). Eating garlic as completely ineffective was considered significantly more often by men than women (32.0%, *N* = 258 for women vs. 49.7%, *N* = 71 for men) (Figure 21). Eating garlic is considered completely ineffective in preventing influenza virus infection by the youngest respondents most often (44.3%, *N* = 98) and least often by respondents 41–60 years old (27.4%, *N* = 64 for 41–50 years old; 28.5%, *N* = 63 for 51–60 years old). This age group is also the most likely to consider eating garlic as completely effective in preventing influenza (9.4%, *N* = 22 for 41–50 years; 8.6%, *N* = 19 for 51–60 years) (*p* = 0.014) (Figure 22). Eating garlic is considered completely ineffective in preventing influenza virus infection by 37.5% (*N* = 198) of pediatric hospital employees and 31% (*N* = 131) of adult hospital employees. One in 11 adult hospital workers considers eating garlic completely effective in preventing influenza virus infections (8.8%, *N* = 37) (Figure 23). Eating garlic is considered completely ineffective in preventing influenza infections by one in two physicians (52.9%, *N* = 162), one in four nurses (26.8%, *N* = 122), and nearly one in four others (23.9%, *N* = 45). One in ten others (9.6%, *N* = 18) and one in eleven nurses (8.6%, *N* = 39) consider eating garlic to be completely effective in preventing influenza virus infection (*p* < 0.001) (Figure 24) (Figures 36 and 37). Eating garlic is often considered completely ineffective in preventing influenza virus infection by those with 5–20 years of work experience (44.5, *N* = 109). At the same time, one in 12 respondents with a seniority of more than 5 years considers eating garlic to be completely effective in preventing influenza virus infection (*p* < 0.001) (Figure 25).

### 3.8. Method 6: Taking Preparations with Inosine 

The use of inosine preparations to prevent influenza virus infection was rated differently by gender, age, and hospital type (Table 8). Men considered preparations with inosine as ineffective more often than women (33.0%, *N* = 266 for women vs. 43.4%, *N* = 62 for men) (Figure 26). Taking preparations with inosine was considered completely ineffective in preventing influenza illness by an average of four in ten people aged 18–40 and over 60, and one in four people aged 41–60 (40.3%, *N* = 89 for 18–30; 42.2%, *N* = 87 for 31–40; 27.8%, *N* = 65 for 41–50; 26.2%, *N* = 58 for 51–60; 42.6%, *N* = 29 for those over 60) (*p* < 0.001) (Figure 27). Taking preparations with inosine was considered completely ineffective in preventing influenza virus infection slightly more often by those working in a pediatric hospital (37.5%, *N* = 198) than in an adult hospital (30.8%, *N* = 130) (Figure 28). Taking preparations with inosine was considered completely ineffective in preventing influenza virus infection by physicians (54.2%, *N* = 166), as well as by one in four nurses (26.5%, *N* = 121) and one in five others (21.8%, *N* = 41) *(p* < 0.001) most often (Figure 29) (Figures 36 and 37). The higher the seniority, the more frequent the statement that taking preparations with inosine is completely ineffective in preventing influenza illness (Figure 30).

### 3.9. Method 7: Take Vitamin C Daily

Daily vitamin C intake to prevent influenza virus infection was rated differently according to gender, age, and seniority (Table 9). Daily vitamin C intake as completely ineffective was considered significantly more often by men than women (39.2%, *N* = 56 for men vs. 25.9%, *N* = 209 for women) (*p* = 0.002) (Figure 31). Daily vitamin C intake as a completely ineffective way to prevent getting influenza was considered by one in three respondents aged 18–40 and over 60 (35.7%, *N* = 79 for 18–30; 36.4%, *N* = 75 for 31–40; 33.8, *N* = 23 for those over 60) and one in five respondents aged 41–60 (17.9%, *N* = 42 for 41–50; 20.8%, *N* = 46 for 51–60). Daily vitamin C intake was considered completely effective in preventing influenza virus infection by one in seven respondents over 60 (14.7%, *N* = 10) and one in ten respondents aged 41–60 (11.8%, *N* = 26 for 51–60; 10.7%, *N* = 25 for 41–50) (*p* < 0.001) (Figure 32). Daily vitamin C intake was considered a completely ineffective way to prevent getting influenza by respondents from pediatric hospitals slightly more often (31.6%, *N* = 167) than respondents from adult hospitals (23.2%, *N* = 98). One in ten respondents from adult hospitals considered daily vitamin C intake as a means of preventing infection against influenza to be completely effective (10.2%, *N* = 43) (*p* = 0.001) (Figure 33). Taking vitamin C daily to prevent influenza infection was considered completely ineffective by one in two physicians (52.3%, *N* = 160), nearly one in five nurses (18.0%, *N* = 82), and one in eight others (12.2%, *N* = 23). At the same time, one in nine nurses (11.6%, *N* = 53), and one in seven others (13.8%, *N* = 26), and only 1% (*N* = 3) of physicians considered daily vitamin C intake as a completely effective way to prevent getting influenza (*p* < 0.001) (Figure 34, Figure 36 and Figure 37). The higher the seniority, the daily intake of vitamin C as a completely ineffective way to prevent influenza virus infection is considered completely ineffective, while at the same time, one in ten people with seniority greater than 5 years believes that daily vitamin C intake in preventing influenza illness is completely effective (*p* < 0.001) (Figure 35).

**Figure 31 vaccines-11-00066-f031:**
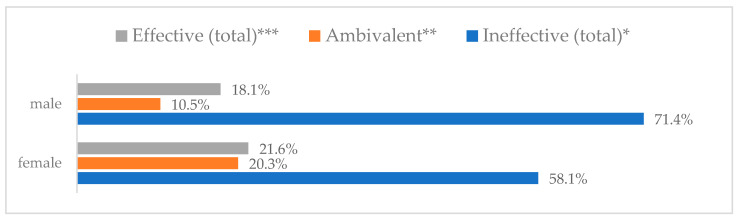
Taking vitamin C daily to prevent influenza virus infection by gender (*N* = 905). * Sum of responses: completely ineffective, ineffective, rather ineffective; ** Response: neither ineffective nor effective; *** Sum of responses: rather effective, effective, completely effective.

**Figure 32 vaccines-11-00066-f032:**
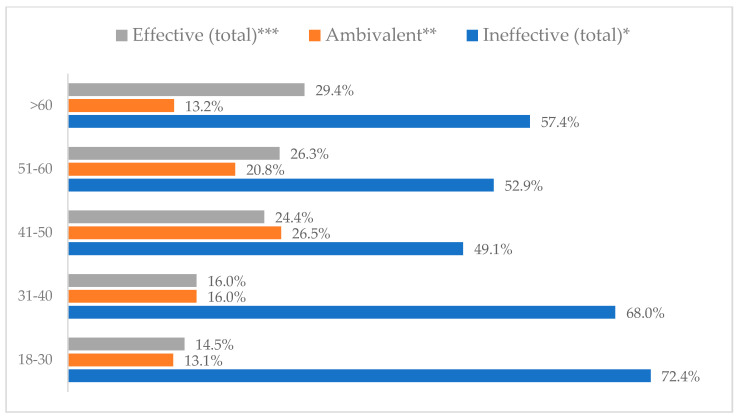
Taking vitamin C daily to prevent influenza virus infection by age (*N* = 905). * Sum of responses: completely ineffective, ineffective, rather ineffective; ** Response: neither ineffective nor effective; *** Sum of responses: rather effective, effective, completely effective.

**Figure 33 vaccines-11-00066-f033:**
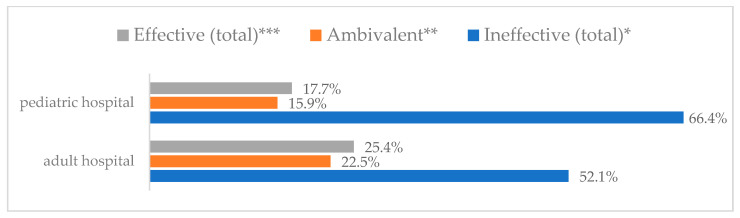
Taking vitamin C daily to prevent influenza virus infection by type of hospital (*N* = 905). * Sum of responses: completely ineffective, ineffective, rather ineffective; ** Response: neither ineffective nor effective; *** Sum of responses: rather effective, effective, completely effective.

**Figure 34 vaccines-11-00066-f034:**
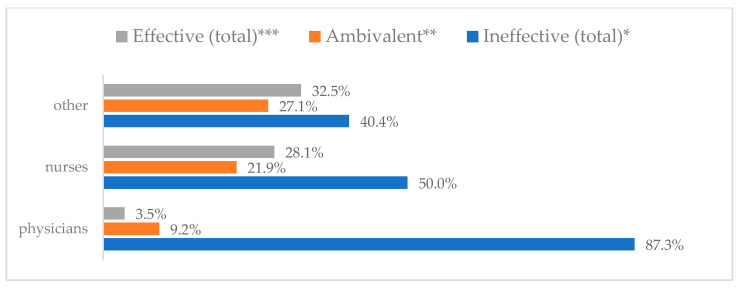
Taking vitamin C daily to prevent influenza virus infection by profession (*N* = 905). * Sum of responses: completely ineffective, ineffective, rather ineffective; ** Response: neither ineffective nor effective; *** Sum of responses: rather effective, effective, completely effective.

**Figure 35 vaccines-11-00066-f035:**
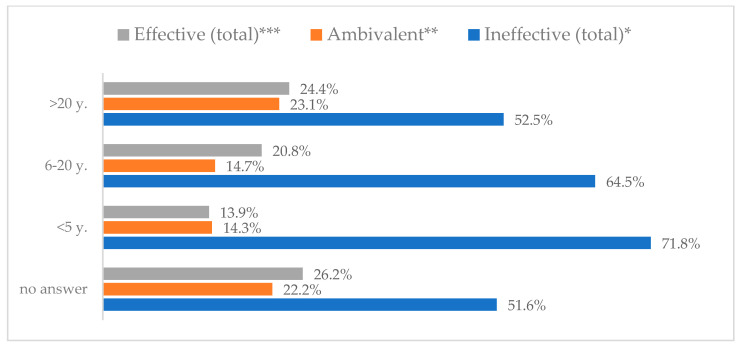
Taking vitamin C daily to prevent influenza virus infection by seniority (*N* = 905). * Sum of responses: completely ineffective, ineffective, rather ineffective; ** Response: neither ineffective nor effective; *** Sum of responses: rather effective, effective, completely effective.

**Figure 36 vaccines-11-00066-f036:**
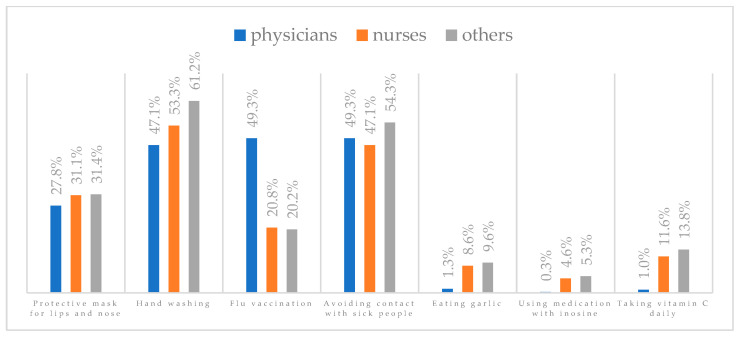
Summary of the indications of each method to prevent influenza virus infection completely effective for each professional group.

**Figure 37 vaccines-11-00066-f037:**
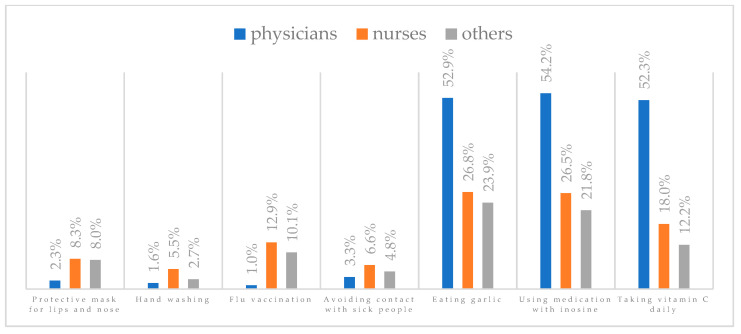
Summary of the indications of each method to prevent influenza virus infection completely ineffective for each professional group.

The findings presented in the second part of the Results relating to eating garlic, taking preparations of inosine, and consuming vitamin C daily to prevent influenza virus infection positively verify hypothesis H2.

**H2.** *Nurses and other hospital staff show greater confidence than physicians in the effectiveness of non-clinical ways to prevent influenza virus infection, such as eating garlic and taking vitamin C daily*.

### 3.10. Attitudes of Vaccinated and Unvaccinated against Influenza Vaccination

One in four respondents (25.3%, *N* = 240) said they had been vaccinated against influenza for the 2019/2020 season. Three-quarters of respondents (73.7%, *N* = 700) said they had not been vaccinated against influenza, and 1.1% (*N* = 10) said they did not remember whether they had been vaccinated. Nearly three times as many people who have been vaccinated against influenza believe that influenza vaccination is a completely effective way to prevent getting influenza (58.8% for vaccinated vs. 20% for unvaccinated) (*p* < 0.001) (Figure 38).

## 4. Discussion

The survey showed significant statistical correlations for all seven ways presented to respondents to prevent influenza virus infection concerning the profession, age, seniority, and type of hospital. 

Hospital employees, especially those with contact with patients, are at high risk of influenza virus infection during the flu season. On the other hand, they may also be carriers of the virus, especially when they become asymptomatic with influenza. For this reason, they were preventing influenza illness, and the spread of the influenza virus in hospitals is significant. Our survey focused on several methods of preventing influenza virus infection. 

Respondents in our survey most often indicated hand washing (52.8%) and avoiding contact with sick people (49.3%) as being completely effective in influenza virus infection. In the case of hand washing, we showed a significant relationship between age, seniority, profession, and the type of hospital where respondents work. Hand washing was considered the least effective by the oldest respondents, significantly more often by nurses than physicians and employees of an adult hospital.

Proper hand washing and using soap or disinfectant prevent the transmission of infectious diseases, reduce the number of nosocomial infections, and improve the quality of healthcare delivery [7]. A randomized trial showed that the people maintaining proper hand hygiene had about 16% fewer upper respiratory tract infections than the control group [8]. Interestingly, the best effect of using canker hygiene was achieved in younger children [9].

While the Wong et al. study did not confirm the effectiveness of hand washing in preventing influenza, it did show that hand hygiene combined with the use of a protective mask significantly reduces the risk of influenza infection [10]. In contrast, a study by Azor-Martinez et al. on a group of college students confirmed the high effectiveness of maintaining proper hand hygiene in interrupting infections during the 2009 swine influenza pandemic [11]. Meanwhile, a case-control study conducted in Fujian Province during seasonal influenza transmission (just before the H1N1 pandemic in China) examined the effect of hand washing on influenza illness. The study’s results indicated that hand washing effectively prevented influenza infection. The authors emphasized that hand washing fulfills its protective function when hand-to-face exposure is limited and secondly when it is one of the hygiene habits [12]. A randomized clinical trial confirms the power of habit concerning hand washing by Larson et al., in which it was shown that single hand washing had a minimal effect on the amount of bacterial flora on hands. However, when hand washing became a hygiene habit, it had a significant effect [13]. In the Liu et al. study, all cases of influenza were laboratory-confirmed and thus refer to influenza, not a mix of other infectious agents [12]. A 2012 Spanish study of patients in hospitals and medical facilities found that frequent hand washing, i.e., more than five times a day, and washing hands after contact with contaminated surfaces significantly reduced influenza virus infection [14]. Even more studies confirm the possibility of influenza A and B virus survival on non-porous surfaces such as steel for as long as 48 h and on porous surfaces such as fabrics or tissues for at least 8–12 h [15]. However, in a cluster randomized controlled trial conducted by Cowling et al. in households, it was found that hand hygiene with or without masks appeared to reduce transmission of the influenza virus. However, there were no significant differences compared to the control group. As a result, hand hygiene and face masks were found to prevent household influenza virus transmission when implemented within 36 h of the onset of symptoms in a family member [16]. 

This study indicated a significant statistical relationship between avoiding contact with sick people to avoid influenza virus infection and demographic categories. Effective avoidance of contact with sick people was declared more often by young people under 30 years of age, pediatric hospital employees, physicians, and other professionals. The WHO and the CDC recommend avoiding contact with sick people to prevent influenza infections [17,18,19], but this is often not possible in a hospital setting, even if it involves other professionals. In our survey, 91.2% (*N* = 866) of respondents reported having contact with patients at work. 

Respondents in our survey further cited a mask covering the mouth and nose (30.1%) and influenza vaccination (29.9%) as effective ways to prevent influenza virus infection. For masks, we showed statistical significance between mask use and respondents’ age, type of hospital, length of service, and profession. For example wearing a mask as ineffective in preventing influenza virus infection was indicated more often by nurses than physicians and those working in an adult hospital for more than five years. 

The surgical mask prevents the spread of influenza infection by limiting hand-to-mouth contact with the person who wears it. When used by medical personnel, it aims to prevent the spread of microorganisms from the wearer to the patient, their advantage being that they are widely available. The literature shows that wearing a mask can significantly reduce influenza virus infection, but certain conditions must be met. In a study by Booth et al., surgical masks were tested to test their effectiveness in preventing influenza infection [20]. The tests showed that a surgical mask could reduce exposure to the infectious influenza virus by an average of six times (much depends on the type of mask). The study’s results indicated that surgical masks, in the context of protection from influenza virus infection, perform their function only to a limited extent. MacIntyre’s study found that masks significantly reduce the risk of influenza-like illness (ILI) infection, but the condition is always wearing them [21]. The significant role of the mask as a barrier to hand-to-face exposure was also described in the study mentioned above by Liu et al. [12]. A meta-analysis by Gralton and McLaws noted that N95 masks have better protection against particles that are similar in size to the influenza virus, with the caveat that the respiratory zone between HCWs and the patient should be extended up to two meters, which is often not possible in hospital work settings. The authors also point out that masks should involve a proper application, wearing, and removal since, especially when removing them, the pathogen can come into contact with the eyes [22]. A systematic review by Xiao et al. confirmed the ineffectiveness of an infected and uninfected person wearing a surgical mask to prevent influenza [23].

In a more recent systematic review, the effectiveness of the surgical mask in preventing influenza infection was rated as low or showed no effect of wearing it in preventing influenza, compared to no mask [24]. Some studies showed no protective effect of a surgical mask, while some showed a significant reduction in infection when wearing a surgical mask compared to no mask [24]. 

Some studies have compared the effectiveness of wearing surgical masks with N95 masks in preventing influenza infection [25]. N95 masks are less available, especially in developing countries. Moreover, they must be sized appropriately to the face and worn tightly to work correctly. Theoretically, N95 masks should be more effective in preventing influenza infection, as they protect against microorganisms spread by droplets and aerosol. However, scientific studies do not support this hypothesis. The effectiveness of these two masks in preventing influenza is similar, probably due to improper selection or wearing of N95 masks, often causing discomfort shortly after donning and resulting in the mask being worn for as short a time as possible [26,27]. 

Based on a meta-analysis by Takahashi et al. that compared the results of studies on the incidence of influenza infections with the use of antiviral mouth and nose masks, it was concluded that there were no statistically significant differences in influenza infection using antiviral masks, which led the authors to conclude that the use of healthcare mouth and nose masks may be insufficient in preventing influenza infections [28]. 

In our study, statistically significant correlations were observed between influenza vaccintion and gender, age, lenghth of service, occupation and type of hospitals where respondents worked. Influenza vaccination in infection prevention was valued more often by employees of pediatric hospitals than of adult hospitals, men and those under 40 years of age. In our survey, most physicians surveyed found influenza vaccination effective in preventing influenza virus infection. Interestingly, the effectiveness of influenza vaccination in preventing influenza virus infection was chosen slightly more often by other hospital employees (including administrative staff) than by nurses. Moreover, one in three nurses surveyed felt that influenza vaccination does not protect against infection. It is a disturbing phenomenon. The influenza vaccine is a seasonal one. Every year, a variety of influenza vaccines are developed around the world, for example, trivalent inactivated, quadrivalent inactivated, trivalent live attenuated, and quadrivalent live attenuated. The effectiveness of influenza vaccination has been repeatedly proven [29,30]. It does not prevent infection 100 percent of the time. However, it reduces the risk of influenza virus infection by about 50 percent, considering seasonal, environmental, and regional variations [17,31,32,33,34].

Similarly low, as in our study, acceptance of vaccination was noted by French researchers. Their 2017–2018 analysis found that only 27% of nurses were vaccinated against influenza [35]. Moreover, in a survey of French nurses conducted a year earlier, influenza vaccination was the most frequently mentioned vaccine with unfavorable opinions, with one in three nurses reporting a declaration of vaccination [36]. In a study by Zhang et al. of British nurses, the influenza vaccination rate was 36%, with four out of ten nurses surveyed having never been vaccinated against influenza [37]. According to an analysis of available studies by Smith et al., even though science provides ample evidence of the effectiveness of influenza vaccination, nurses’ vaccination rates are inadequate [38]. It is a significant problem because, as the analysis shows, reluctance to vaccinate against influenza translates into the advice nurses give patients. As studies in behavioral health care workers indicate, self-vaccination increases the chance they will recommend vaccination to patients [39]. 

Our study also showed significant statistical relationships to preventing influenza by eating garlic, consuming vitamin C daily, and using products with inosine. 

Garlic preparations are taken to prevent or treat cold symptoms, lower cholesterol, or regulate blood pressure [40,41]. The prevalence of taking garlic to prevent infections varies by country, from about 3.5% in the US to 10.7% in Australia [42]. In a survey by Agnete et al. of 2500 people in Norway, Sweden, and the Netherlands during the COVID-19 pandemic, 4.2% of respondents were supplemented with garlic [43]. A Cochrane systematic review by Lissiman et al. evaluating the effectiveness of garlic preparations taken for three months in preventing the common cold confirmed fewer infections in the group taking garlic compared to the placebo. However, the review’s authors emphasize that the studies sponsored by supplement manufacturers, in which the efficacy of garlic was unconfirmed, may have needed to be published and thus were not included in the review [44]. In a randomized, double-blind study by Nantz et al., there were no differences in the incidence of colds and influenza between the group taking a placebo and the group that took a capsule extract of aged garlic for 90 days [45]. 

The results of previous studies suggest no or little effect of vitamin C on shortening the duration of the common cold regardless of the dose taken [46]. The same meta-analysis confirmed that daily prophylactic intake of Vitamin C at a dose of 0.2 g/day does not reduce the frequency of respiratory tract infections. In contrast, a more recent systematic review by Yuan et al. unequivocally concluded that current high-quality scientific evidence does not support the efficacy of vitamin C intake for influenza prevention [47]. 

Another potential immunostimulant we studied was inosine pranobex, a popular over-the-counter preparation advertised as an effective antiviral. The immunomodulatory effects of inosine have only been confirmed in vivo studies. In a study conducted in the Czech Republic and Slovakia among patients with ILI, it was shown that the difference in time to resolution of flu-like symptoms between the group of patients who received inosine pranobex and the group of patients who received placebo was not statistically significant. However, at the same time, faster improvement in health was shown in those in the group receiving inosine pranobex compared to those in the placebo group [48]. For this reason, it is not recommended by scientific societies as an agent to increase immunity against influenza or as a treatment for influenza [49].

### Study Limitations

Our study has some limitations. The study population is a partial reflection of the population of hospital employees, which affects the generalizability of the study results. Moreover, the study was conducted in academic hospitals where physicians and nurses teach students. Therefore, it is reasonable to assume that our respondents’ knowledge of infection prevention, based on experience and research findings, was higher than that of HWs in other hospitals. For this reason, it would be worthwhile to conduct a comparative study in other non-academic hospitals to examine the attitudes of physicians and nurses toward the prevention of influenza infection.

## 5. Conclusions and Practical Considerations

In our survey, we examined what methods of influenza prevention healthcare workers believe in. According to the survey, most respondents have confidence in methods that can effectively prevent influenza, and this belief is more robust among physicians than nurses. Washing hands was considered most effective in preventing influenza, followed by using a protective mask, and in third place by vaccination. We showed that pediatric hospital staff considers influenza vaccination more effective in preventing influenza than adult hospital staff. We confirmed that some medical professionals believe in “natural” methods of influenza prevention, such as daily consumption of vitamin C or the use of inosine pranobex. 

We hope that the results of our study bring important information for planning the promotion of influenza vaccination among healthcare workers in Poland and EU. 

## Figures and Tables

**Figure 1 vaccines-11-00066-f001:**
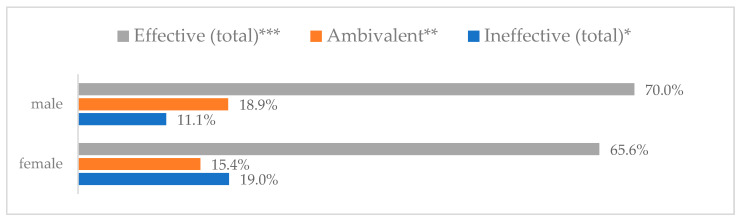
Wearing a protective mask that covers the mouth and nose to prevent influenza virus infection by gender (*N* = 905). * Sum of responses: completely ineffective, ineffective, rather ineffective; ** Response: neither ineffective nor effective; *** Sum of responses: rather effective, effective, completely effective.

**Figure 2 vaccines-11-00066-f002:**
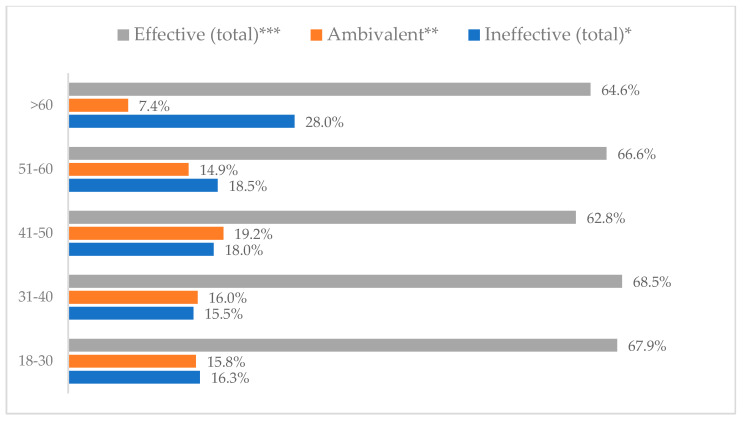
Wearing a protective mask that covers the mouth and nose to prevent influenza virus infection by age (*N* = 905). * Sum of responses: completely ineffective, ineffective, rather ineffective; ** Response: neither ineffective nor effective; *** Sum of responses: rather effective, effective, completely effective.

**Figure 3 vaccines-11-00066-f003:**
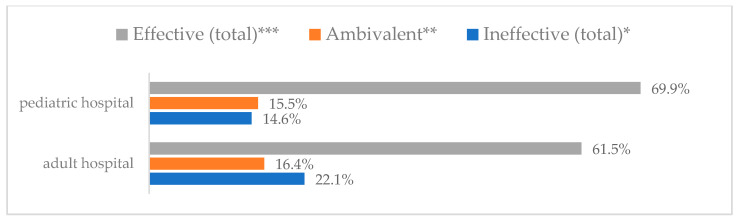
Wearing a protective mask that covers the mouth and nose to prevent influenza virus infection by type of hospital (*N* = 905). * Sum of responses: completely ineffective, ineffective, rather ineffective; ** Response: neither ineffective nor effective; *** Sum of responses: rather effective, effective, completely effective.

**Figure 4 vaccines-11-00066-f004:**
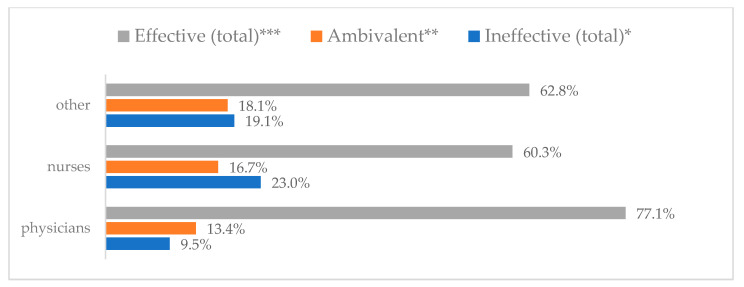
Wearing a protective mask that covers the mouth and nose to prevent influenza virus infection by profession (*N* = 905). * Sum of responses: completely ineffective, ineffective, rather ineffective; ** Response: neither ineffective nor effective; *** Sum of responses: rather effective, effective, completely effective.

**Figure 5 vaccines-11-00066-f005:**
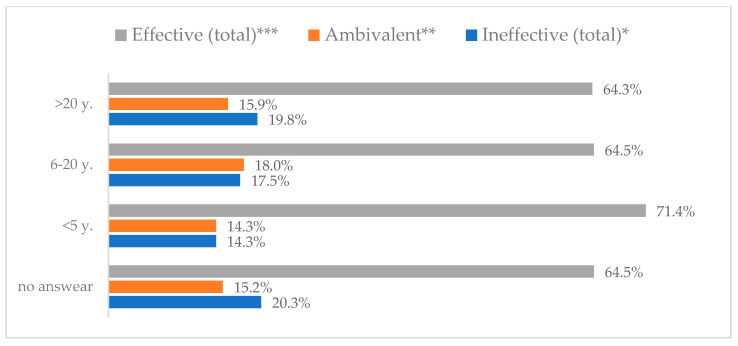
Wearing a protective mask that covers the mouth and nose to prevent influenza virus infection by seniority (*N* = 905). * Sum of responses: completely ineffective, ineffective, rather ineffective; ** Response: neither ineffective nor effective; *** Sum of responses: rather effective, effective, completely effective.

**Figure 6 vaccines-11-00066-f006:**
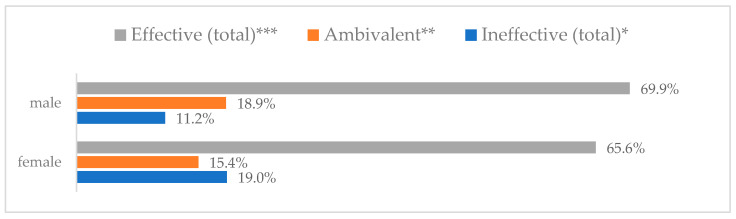
Hand washing to prevent influenza virus infection by gender (*N* = 905). * Sum of responses: completely ineffective, ineffective, rather ineffective; ** Response: neither ineffective nor effective; *** Sum of responses: rather effective, effective, completely effective.

**Figure 7 vaccines-11-00066-f007:**
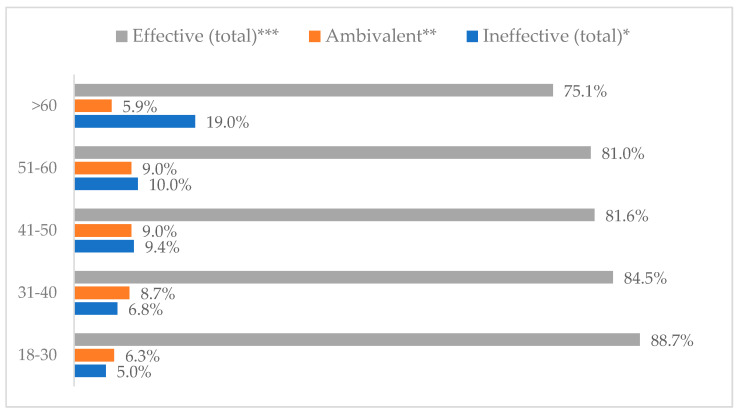
Hand washing to prevent influenza virus infection by age (*N* = 905). * Sum of responses: completely ineffective, ineffective, rather ineffective; ** Response: neither ineffective nor effective; *** Sum of responses: rather effective, effective, completely effective.

**Figure 8 vaccines-11-00066-f008:**
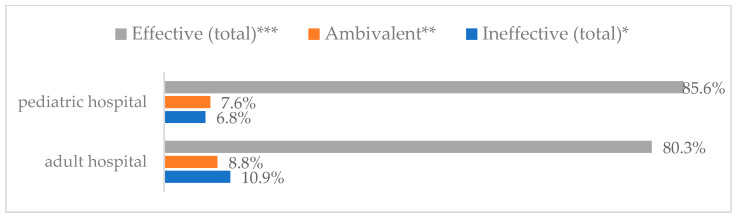
Hand washing as a means of preventing influenza virus infection by type of hospital (*N* = 905). * Sum of responses: completely ineffective, ineffective, rather ineffective; ** Response: neither ineffective nor effective; *** Sum of responses: rather effective, effective, completely effective.

**Figure 9 vaccines-11-00066-f009:**
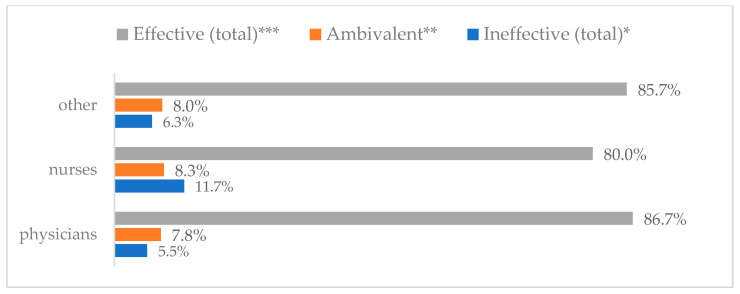
Hand washing to prevent influenza virus infection by profession (*N* = 905). * Sum of responses: completely ineffective, ineffective, rather ineffective; ** Response: neither ineffective nor effective; *** Sum of responses: rather effective, effective, completely effective.

**Figure 10 vaccines-11-00066-f010:**
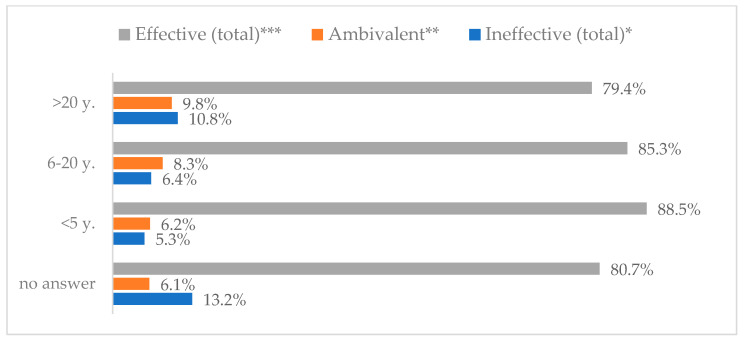
Hand washing to prevent influenza virus infection by seniority (*N* = 905). * Sum of responses: completely ineffective, ineffective, rather ineffective; ** Response: neither ineffective nor effective; *** Sum of responses: rather effective, effective, completely effective.

**Figure 11 vaccines-11-00066-f011:**
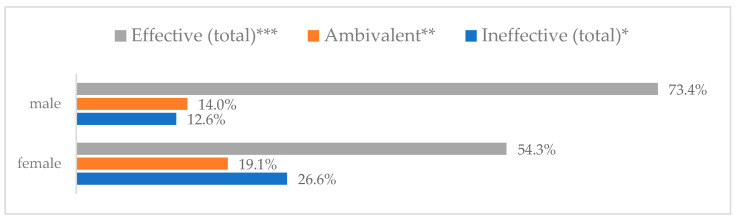
Influenza vaccination to prevent influenza virus infection by gender (*N* = 905). * Sum of responses: completely ineffective, ineffective, rather ineffective; ** Response: neither ineffective nor effective; *** Sum of responses: rather effective, effective, completely effective.

**Figure 12 vaccines-11-00066-f012:**
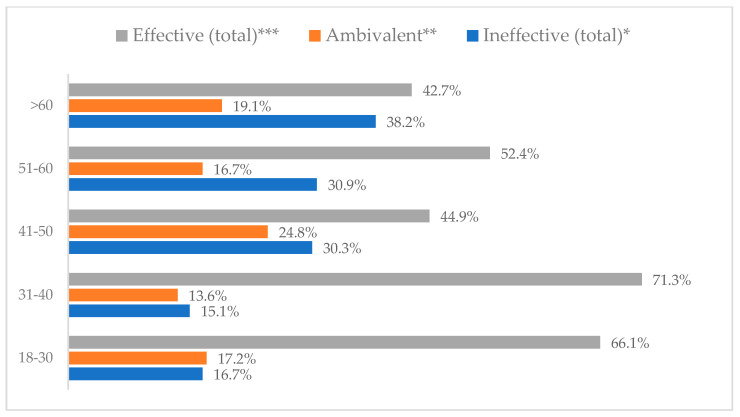
Influenza vaccination to prevent influenza virus infection by age (*N* = 905). * Sum of responses: completely ineffective, ineffective, rather ineffective; ** Response: neither ineffective nor effective; *** Sum of responses: rather effective, effective, completely effective.

**Figure 13 vaccines-11-00066-f013:**
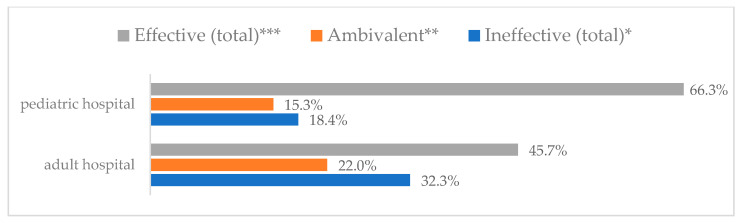
Influenza vaccination to prevent influenza virus infection by type of hospital (*N* = 905). * Sum of responses: completely ineffective, ineffective, rather ineffective; ** Response: neither ineffective nor effective; *** Sum of responses: rather effective, effective, completely effective.

**Figure 14 vaccines-11-00066-f014:**
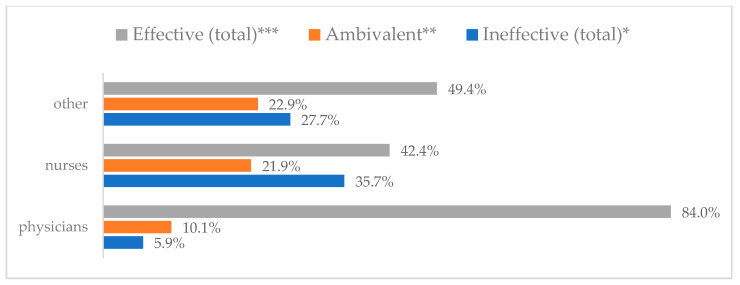
Influenza vaccination to prevent influenza virus infection by profession (*N* = 905). * Sum of responses: completely ineffective, ineffective, rather ineffective; ** Response: neither ineffective nor effective; *** Sum of responses: rather effective, effective, completely effective.

**Figure 15 vaccines-11-00066-f015:**
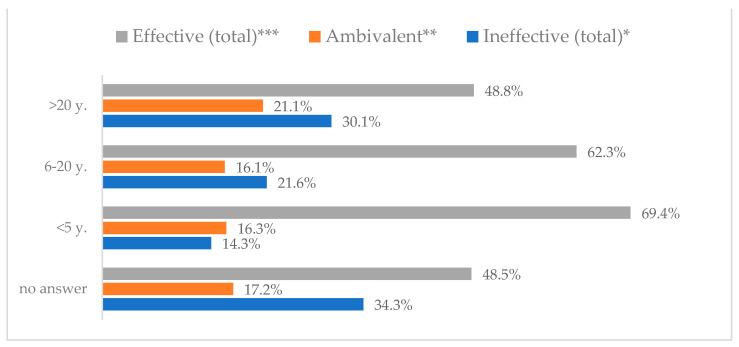
Influenza vaccination to prevent influenza virus infection by seniority (*N* = 905). * Sum of responses: completely ineffective, ineffective, rather ineffective; ** Response: neither ineffective nor effective; *** Sum of responses: rather effective, effective, completely effective.

**Figure 16 vaccines-11-00066-f016:**
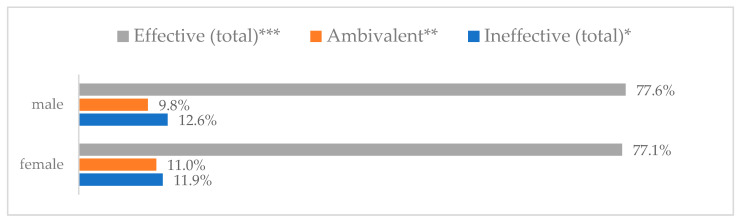
Avoiding contact with sick people to prevent influenza virus infection by gender (*N* = 905). * Sum of responses: completely ineffective, ineffective, rather ineffective; ** Response: neither ineffective nor effective; *** Sum of responses: rather effective, effective, completely effective.

**Figure 17 vaccines-11-00066-f017:**
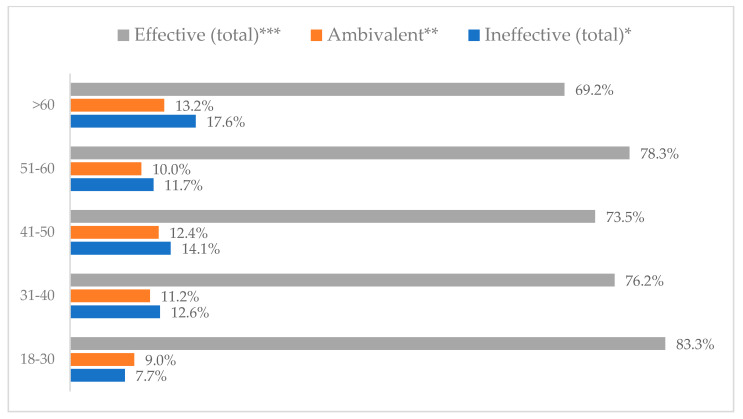
Avoiding contact with sick people to prevent influenza virus infection by age (*N* = 905). * Sum of responses: completely ineffective, ineffective, rather ineffective; ** Response: neither ineffective nor effective; *** Sum of responses: rather effective, effective, completely effective.

**Figure 18 vaccines-11-00066-f018:**
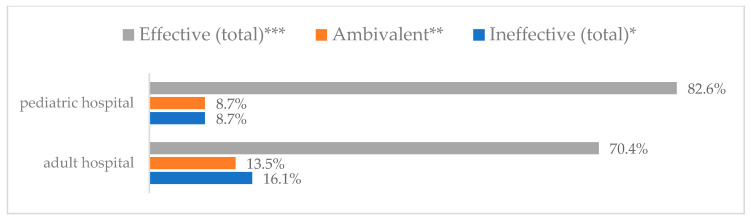
Avoiding contact with sick people to prevent influenza virus infection by type of hospital (*N* = 905). * Sum of responses: completely ineffective, ineffective, rather ineffective; ** Response: neither ineffective nor effective; *** Sum of responses: rather effective, effective, completely effective.

**Figure 19 vaccines-11-00066-f019:**
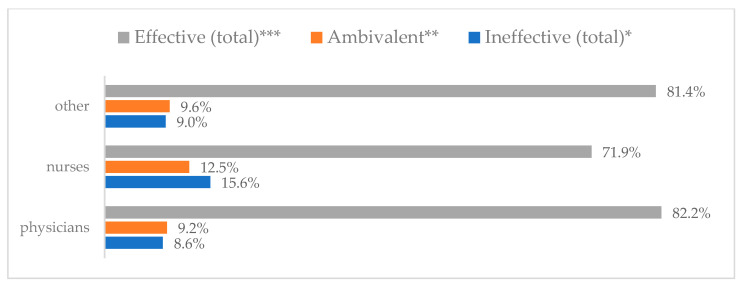
Avoiding contact with sick people to prevent influenza virus infection by profession (*N* = 905). * Sum of responses: completely ineffective, ineffective, rather ineffective; ** Response: neither ineffective nor effective; *** Sum of responses: rather effective, effective, completely effective.

**Figure 20 vaccines-11-00066-f020:**
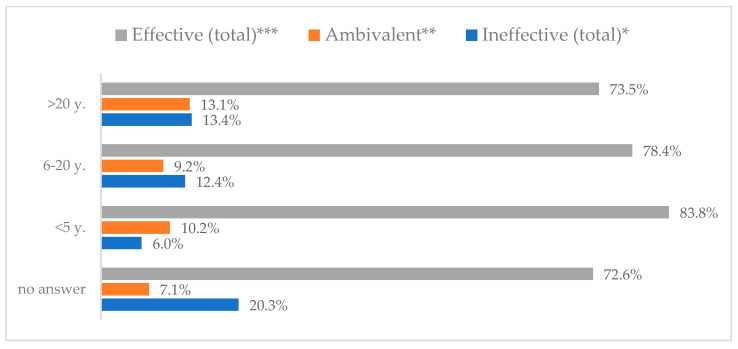
Avoiding contact with sick people to prevent influenza virus infection by seniority (*N* = 905). * Sum of responses: completely ineffective, ineffective, rather ineffective; ** Response: neither ineffective nor effective; *** Sum of responses: rather effective, effective, completely effective.

**Figure 21 vaccines-11-00066-f021:**
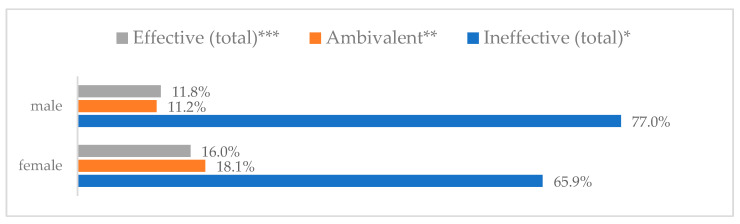
Eating garlic to prevent influenza virus infection by gender (*N* = 905). * Sum of responses: completely ineffective, ineffective, rather ineffective; ** Response: neither ineffective nor effective; *** Sum of responses: rather effective, effective, completely effective.

**Figure 22 vaccines-11-00066-f022:**
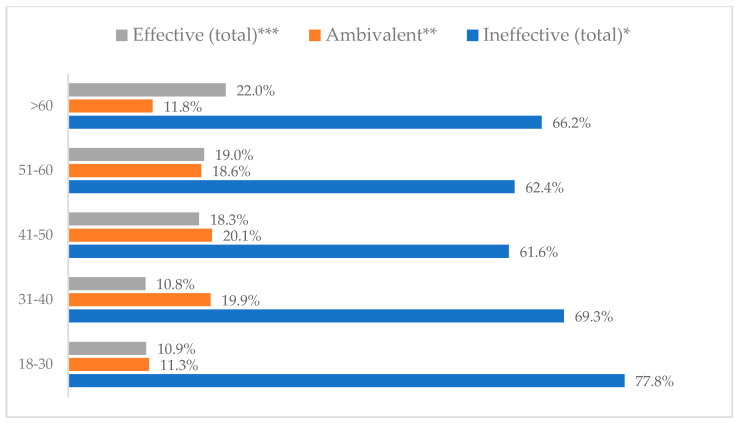
Eating garlic to prevent influenza virus infection by age (*N* = 905). * Sum of responses: completely ineffective, ineffective, rather ineffective; ** Response: neither ineffective nor effective; *** Sum of responses: rather effective, effective, completely effective.

**Figure 23 vaccines-11-00066-f023:**
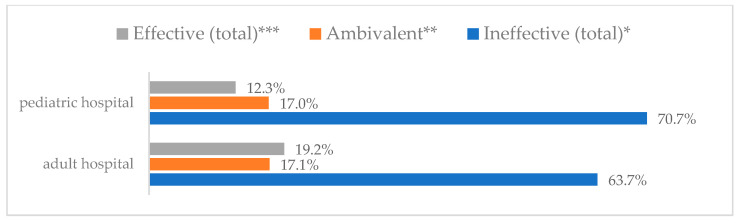
Eating garlic to prevent influenza virus infection by type of hospital (*N* = 905). * Sum of responses: completely ineffective, ineffective, rather ineffective; ** Response: neither ineffective nor effective; *** Sum of responses: rather effective, effective, completely effective.

**Figure 24 vaccines-11-00066-f024:**
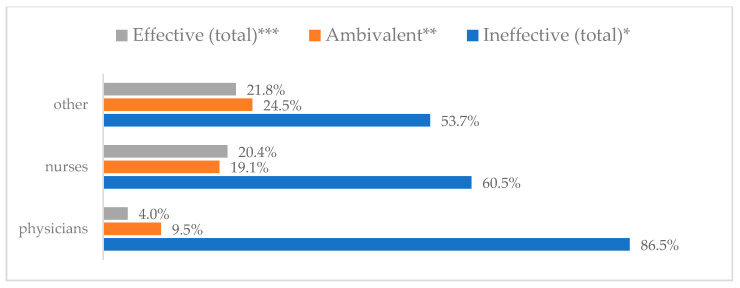
Eating garlic to prevent influenza virus infection by profession (*N* = 905). * Sum of responses: completely ineffective, ineffective, rather ineffective; ** Response: neither ineffective nor effective; *** Sum of responses: rather effective, effective, completely effective.

**Figure 25 vaccines-11-00066-f025:**
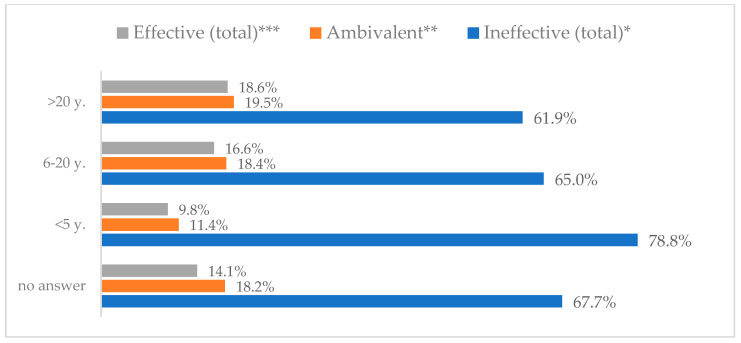
Eating garlic to prevent influenza virus infection by seniority (*N* = 905). * Sum of responses: completely ineffective, ineffective, rather ineffective; ** Response: neither ineffective nor effective; *** Sum of responses: rather effective, effective, completely effective.

**Figure 26 vaccines-11-00066-f026:**
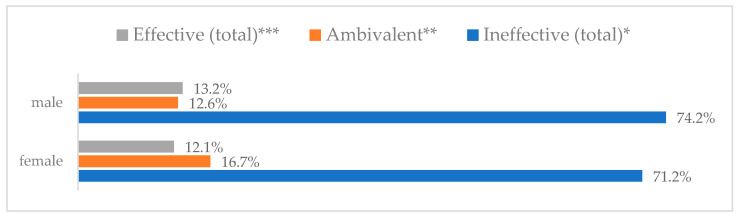
Taking preparations with inosine to prevent influenza virus infection by gender (*N* = 905). * Sum of responses: completely ineffective, ineffective, rather ineffective; ** Response: neither ineffective nor effective; *** Sum of responses: rather effective, effective, completely effective.

**Figure 27 vaccines-11-00066-f027:**
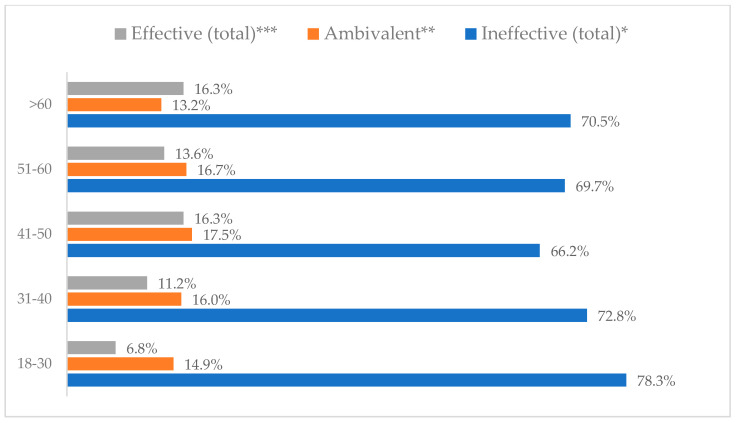
Taking preparations with inosine to prevent influenza virus infection by age (*N* = 905). * Sum of responses: completely ineffective, ineffective, rather ineffective; ** Response: neither ineffective nor effective; *** Sum of responses: rather effective, effective, completely effective.

**Figure 28 vaccines-11-00066-f028:**
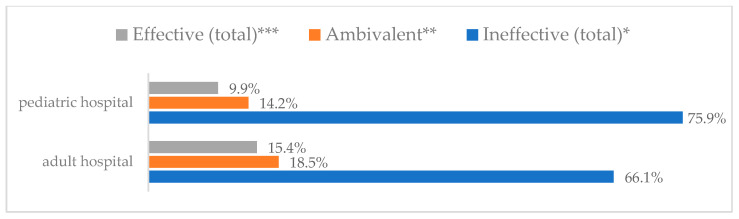
Taking preparations with inosine to prevent influenza virus infection by type of hospital (*N* = 905). * Sum of responses: completely ineffective, ineffective, rather ineffective; ** Response: neither ineffective nor effective; *** Sum of responses: rather effective, effective, completely effective.

**Figure 29 vaccines-11-00066-f029:**
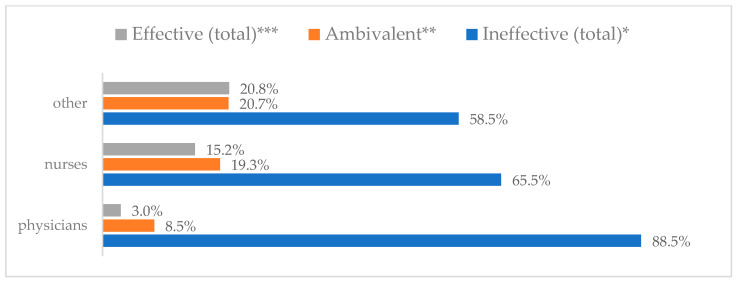
Taking preparations with inosine to prevent influenza virus infection by profession (*N* = 905). * Sum of responses: completely ineffective, ineffective, rather ineffective; ** Response: neither ineffective nor effective; *** Sum of responses: rather effective, effective, completely effective.

**Figure 30 vaccines-11-00066-f030:**
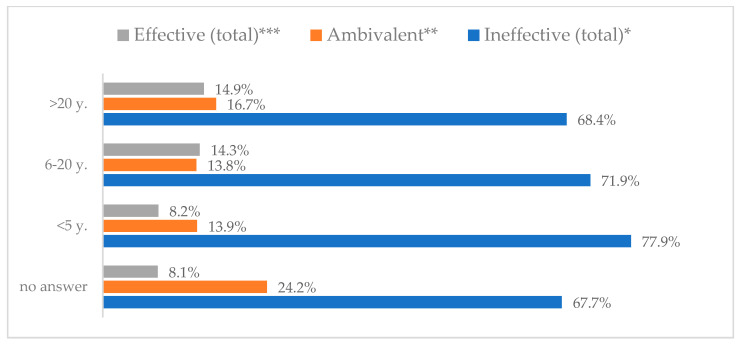
Taking preparations with inosine to prevent influenza virus infection by seniority (*N* = 905). * Sum of responses: completely ineffective, ineffective, rather ineffective; ** Response: neither ineffective nor effective; *** Sum of responses: rather effective, effective, completely effective.

**Figure 38 vaccines-11-00066-f038:**
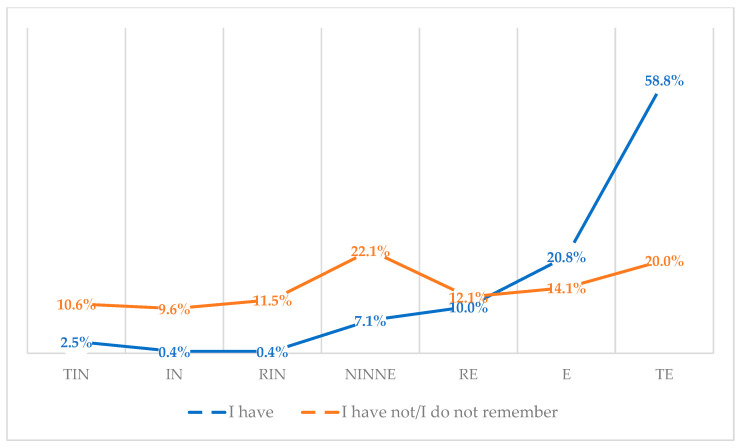
Attitudes toward influenza vaccination to prevent influenza virus infection among vaccinated and unvaccinated against influenza (*N* = 950). Note. TIN-totally ineffective, IN-ineffective, RIN-rather ineffective, NINE-neither ineffective nor effective, RE-rather effective, E-effective, TE-totally effective.

**Table 1 vaccines-11-00066-t001:** Sociodemographic characteristics (*N* = 950).

Variables	N (%)
Gender	
Female	807 (84.9)
Male	143 (15.1)
Age	
18–30	221 (23.3)
31–40	206 (21.7)
41–50	234 (24.6)
51–60	221 (23.3)
>60	68 (7.1)
Type of hospital	
Adult hospital	422 (44.4)
Pediatric hospital	528 (55.6)
Profession	
Physician	306 (32.2)
Nurse	456 (48.0)
Other	188 (19.8)
Seniority	
<5 y.	245 (25.8)
6–20 y.	217 (22.8)
>20 y.	389 (40.9)
Refuse to answer	99 (10.5)

**Table 2 vaccines-11-00066-t002:** Evaluation of the effectiveness of methods to prevent influenza infections (*N* = 950).

Mean	SE	MD	OR	Percentiles			The Indicated Degree of Effectiveness of the Method to Prevent Influenza Virus Infection *
				25th	50th	75th	
Mask covering mouth and nose							
5.08	0.058	5.00	1.796	4.00	5.00	7.00	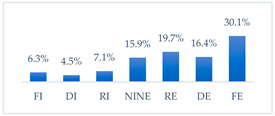
Hand washing							
5.88	0.051	7.00	1.564	5.00	7.00	7.00	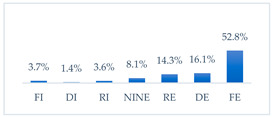
Influenza vaccination							
4.84	0.064	5.00	1.970	4.00	5.00	7.00	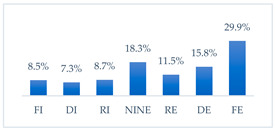
Avoiding contact with sick people							
5.67	0.056	6.00	1.731	5.00	6.00	7.00	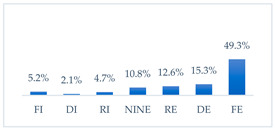
Eating garlic							
2.77	0.057	2.00	1.769	1.00	2.00	4.00	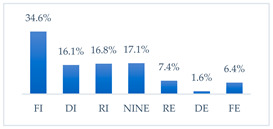
Taking preparations with inosine							
2.61	0.052	2.00	1.610	1.00	2.00	4.00	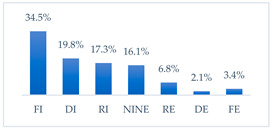
Taking vitamin C daily							
3.12	0.061	3.00	1.888	1.00	3.00	4.00	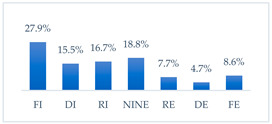

* FI—fully ineffective, DI—definitely ineffective, RI—rather ineffective, NINE—neither ineffective nor effective, RE—rather effective, DE—definitely effective, FE—fully effective.

**Table 3 vaccines-11-00066-t003:** Evaluation of mouth and nose covering mask to prevent influenza virus infection by sociodemographic category (*N* = 950).

Variables	Mean	95% CI	SE	MD	OR	χ^2^	df	*p*-Value
Gender								
Female	5.05	4.92–5.18	0.065	5.00	1.836	8.251	6	0.220
Male	5.23	4.98–5.49	0.129	5.00	1.546			
Age								
18–30	5.05	4.83–5.26	0.111	5.00	1.645	41.541	24	0.015
31–40	5.11	4.87–5.35	0.112	5.00	1.748			
41–50	5.06	4.83–5.29	0.118	5.00	1.803			
51–60	5.16	4.91–5.42	0.125	5.00	1.863			
>60	4.88	4.36–5.41	0.262	5.00	2.162			
Type of hospital								
Adult hospital	4.93	4.75–5.12	0.093	5.00	1.908	13.162	6	0.041
Pediatric hospital	5.19	5.05–5.34	0.074	5.00	1.694			
Profession								
Physicians	5.43	5.27–5.59	0.082	6.00	1.443	53.776	12	<0.001
Nurses	4.88	4.70–5.06	0.091	5.00	1.939			
Others	4.99	4.72–5.26	0.137	5.00	1.876			
Seniority								
<5 y.	5.19	4.98–5.40	0.481	6.00	1.649	38.867	18	0.003
6–20 y.	5.01	4.77–5.24	0.120	5.00	1.761			
>20 y.	5.10	4.91–5.29	0.096	5.00	1.903			
Refuse to answer	4.87	4.51–5.23	0.180	5.00	1.788			

**Table 4 vaccines-11-00066-t004:** Evaluation of handwashing as a means of preventing influenza virus infection by sociodemographic category (*N* = 950).

Variables	Mean	95% CI	SE	MD	OR	χ^2^	df	*p*-Value
Gender								
Female	5.89	5.78–6.00	0.055	7.00	1.562	6.759	6	0.334
Male	5.80	5.54–6.06	0.132	6.00	1.577			
Age								
18–30	5.99	5.82–6.15	0.084	6.00	1.249	76.026	24	<0.001
31–40	5.87	5.67–6.07	0.101	6.00	1.456			
41–50	5.89	5.68–6.10	0.337	7.00	1.645			
51–60	5.86	5.64–6.08	0.112	7.00	1.660			
>60	5.51	4.99–6.03	0.260	7.00	2.147			
Type of hospital								
Adult hospital	5.78	5.61–5.95	0.083	7.00	1.748	22.752	6	<0.001
Pediatric hospital	5.95	5.84–6.07	0.061	7.00	1.396			
Profession								
Physicians	5.94	5.79–6.10	0.077	6.00	1.340	33.346	12	<0.001
Nurses	5.75	5.59–5.91	0.081	7.00	1.731			
Others	6.07	5.87–6.28	0.106	7.00	1.450			
Seniority								
<5 y.	6.00	5.84–6.16	0.083	6.00	1.293	66.493	18	<0.001
6–20 y.	5.89	5.70–6.09	0.097	7.00	1.428			
>20 y.	5.84	5.67–6.01	0.088	7.00	1.740			
Refuse to answer	5.68	5.33–6.02	0.173	6.00	1.719			

**Table 5 vaccines-11-00066-t005:** Evaluation of influenza vaccination as a means of preventing influenza virus infection by sociodemographic category (*N* = 950).

Variables	Mean	95% CI	SE	MD	OR	χ^2^	df	*p*-Value
Gender								
Female	4.72	4.58–4.86	0.070	5.00	1.991	20.859	6	0.002
Male	5.51	5.23–5.79	0.143	6.00	1.707			
Age								
18–30	5.23	5.00–5.47	0.119	6.00	1.775	82.197	24	<0.001
31–40	5.32	5.07–5.57	0.125	6.00	1.787			
41–50	4.36	4.11–4.61	0.128	4.00	1.952			
51–60	4.66	4.38–4.93	0.141	5.00	2.093			
>60	4.35	3.82–4.89	0.267	4.00	2.204			
Type of hospital								
Adult hospital	4.39	4.20–4.58	0.098	4.00	2.007	58.481	6	<0.001
Pediatric hospital	5.20	5.04–5.36	0.081	6.00	1.866			
Profession								
Physicians	5.97	5.82–6.12	0.077	6.00	1.353	165.902	12	<0.001
Nurses	4.23	4.04–4.41	0.095	4.00	2.020			
Others	4.48	4.21–4.54	0.139	4.00	1.911			
Seniority								
<5 y.	5.37	5.15–5.59	0.111	6.00	1.740	62.899	12	<0.001
6–20 y.	4.96	4.70–5.21	0.130	5.00	1.908			
**>20 y.**	4.56	4.36–4.77	0.104	4.00	2.058			

**Table 6 vaccines-11-00066-t006:** Rating of avoiding contact with sick people to prevent influenza virus infection by sociodemographic category (*N* = 950).

Variables	Mean	95% CI	SE	MD	OR	χ^2^	df	*p*-Value
Gender								
Female	5.68	5.56–5.80	0.061	7.00	1.719	6.750	6	0.345
Male	5.57	5.27–5.86	0.151	6.00	1.802			
Age								
18–30	5.89	5.70–6.08	0.097	6.00	1.446	41.724	24	0.014
31–40	5.61	5.37–5.84	0.120	6.00	1.729			
41–50	5.52	5.28–5.75	0.120	6.00	1.837			
51–60	5.75	5.52–5.98	0.116	7.00	1.731			
>60	5.35	4.84–5.86	0.256	6.50	2.114			
Type of hospital								
Adult hospital	5.39	5.21–5.57	0.092	6.00	1.889	24.783	6	<0.001
Pediatric hospital	5.88	5.75–6.02	0.068	7.00	1.561			
Profession								
Physicians	5.84	5.66–6.01	0.089	6.00	1.549	22.939	12	0.028
Nurses	5.46	5.29–5.64	0.087	6.00	1.864			
Others	5.87	5.64–6.11	0.119	7.00	1.627			
Seniority								
<5 y.	5.92	5.74–6.10	0.091	6.00	1.428	42.802	18	<0.001
6–20 y.	5.72	5.50–5.95	0.116	7.00	1.702			
>20 y.	5.54	5.36–5.71	0.094	6.00	1.853			
Refuse to answer	5.39	5.01–5.77	0.192	6.00	1.910			

**Table 7 vaccines-11-00066-t007:** Evaluation of eating garlic to prevent influenza virus infection by sociodemographic category (*N* = 950).

Variables	Mean	95% CI	SE	MD	OR	χ^2^	df	*p*-Value
Gender								
Female	2.85	2.73–2.97	0.062	3.00	1.770	19.857	6	0.003
Male	2.30	2.02–2.58	0.142	2.00	1.695			
Age								
18–30	2.37	2.15–2.58	0.110	2.00	1.631	41.606	24	0.014
31–40	2.56	2.34–2.78	0.111	2.00	1.591			
41–50	3.08	2.84–3.32	0.121	3.00	1.851			
51–60	3.01	2.77–3.25	0.123	3.00	1.829			
>60	2.87	2.40–3.34	0.235	2.00	1.939			
Type of hospital								
Adult hospital	2.95	2.77–3.13	0.091	3.00	1.869	12.990	6	0.043
Pediatric hospital	2.62	2.48–2.76	0.073	2.00	1.673			
Profession								
Physicians	1.98	1.84–2.13	0.074	1.00	1.294	102.456	12	<0.001
Nurses	3.10	2.93–3.27	0.086	3.00	1.840			
Others	3.25	2.98–3.52	0.135	3.00	1.849			
Seniority								
<5 y.	2.31	2.11–2.50	0.099	2.00	1.542	33.377	18	0.015
6–20 y.	2.84	2.59–3.09	0.126	3.00	1.855			
>20 y.	3.00	2.82–3.19	0.093	3.00	1.842			
Refuse to answer	2.84	2.52–3.16	0.162	3.00	1.614			

**Table 8 vaccines-11-00066-t008:** Taking preparations with inosine to prevent influenza virus infection by sociodemographic category (*N* = 950).

Variables	Mean	95% CI	SE	MD	OR	χ^2^	df	*p*-Value
Gender								
Female	2.64	2.53–2.75	0.056	2.00	1.592	7.795	6	0.254
Male	2.45	2.17–2.73	0.142	2.00	1.702			
Age								
18–30	2.30	2.11–2.49	0.097	2.00	1.444	57.539	24	<0.001
31–40	2.43	2.21–2.64	0.110	2.00	1.578			
41–50	2.86	2.66–3.07	0.105	3.00	1.599			
51–60	2.80	2.58–3.02	0.112	2.00	1.659			
>60	2.65	2.19–3.10	0.228	2.00	1.883			
Type of hospital								
Adult hospital	2.80	2.64–2.96	0.082	3.00	1.675	13.356	6	0.038
Pediatric hospital	2.46	2.32–2.59	0.067	2.00	1.541			
Profession								
Physicians	1.88	1.74–2.01	0.067	1.00	1.175	112.718	12	<0.001
Nurses	2.87	2.72–3.03	0.078	3.00	1.665			
Others	3.15	2.91–3.40	0.122	3.00	1.675			
Seniority								
<5 y.	2.30	2.12–2.48	0.092	2.00	1.433	26.390	18	0.091
6–20 y.	2.59	2.37–2.82	0.114	2.00	1.681			
>20 y.	2.80	2.63–2.97	0.085	3.00	1.673			
Refuse to answer	2.65	2.35–2.95	0.151	2.00	1.507			

**Table 9 vaccines-11-00066-t009:** Taking vitamin C daily to prevent influenza virus infection by sociodemographic category (*N* = 950).

Variables	Mean	95% CI	SE	MD	OR	χ^2^	df	*p*-Value
Gender								
Female	3.19	3.06–3.32	0.066	3.00	1.874	20.548	6	0.002
Male	2.69	2.38–3.01	0.160	2.00	1.918			
Age								
18–30	2.60	2.38–2.82	0.113	2.00	1.683	74.040	24	<0.001
31–40	2.73	2.49–2.98	0.125	2.00	1.800			
41–50	3.56	3.32–3.79	0.110	4.00	1.825			
51–60	3.49	3.24–3.75	0.131	3.00	1.914			
>60	3.22	2.69–3.75	0.267	2.50	2.198			
Type of hospital								
Adult hospital	3.41	3.22–3.59	0.093	3.00	1.920	22.345	6	0.001
Pediatric hospital	2.89	2.73–3.04	0.080	3.00	1.831			
Profession								
Physicians	1.93	1.79–2.07	0.072	1.00	1.254	205.275	12	<0.001
Nurses	3.59	3.42–3.77	0.089	3.50	1.892			
Others	3.89	3.62–4.15	0.134	4.00	1.834			
Seniority								
<5 y.	2.55	2.35–2.76	0.104	2.00	1.628	62.446	18	<0.001
6–20 y.	3.00	2.74–3.26	0.134	3.00	1.967			
>20 y.	3.43	3.24–3.63	0.098	3.00	1.926			
Refuse to answer	3.53	3.16–3.89	0.182	3.00	1.815			

## Data Availability

The data presented in this study are available on request from the corresponding author.

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
