# Peer review of "Vaccine or Garlic–Is It a Choice? Awareness of Medical Personnel on Prevention of Influenza Infections"

_vaccines, 2022, doi:10.3390/vaccines11010066_

Round 1

Reviewer 1 Report

This is a nice study of an important topic with a good review of the evidence. Potential limitations are clearly detailed.

Several minor comments.

Figure 10 explain the abbreviations 'tak' and 'nie'

Moderate English language editing is needed. The term 'washing cancers' probably needs a better English term.

Author Response

This is a nice study of an important topic with a good review of the evidence. Potential limitations are clearly detailed.

Thank you very much for reviewing the article and the comments indicated. We have taken all of them into consideration.

Several minor comments.

Figure 10 explain the abbreviations 'tak' and 'nie'

We have completed the description of the table.

Moderate English language editing is needed. The term 'washing cancers' probably needs a better English term.

We conducted a thorough linguistic analysis of the article.

Reviewer 2 Report

Thank you for the invitation. This manuscript is timely and interesting but lacks rigor in the methods and presentation of the results. I have few comments below;

Abstract: This section does not provide quantitative results. The authors are encouraged to provide the details of major results. Please remove the numbering before each heading. The method section also needs further elaboration.

Title:

Is it necessary to put Garlic in the title? As I have seen that the authors have evaluated various methods in this research, but only garlic was mentioned in the title? 

introduction

The rationale of this study is not clear. Is there any study on the similar topic in Poland or any other European country? If not, please state in this section. However, the authors are required to demonstrate why this study is needed and what new information is added to the existing local and international literature. 

Methods:

The sample size heading does not provide any information on how the sample size was estimated.

It is not clear whether the questionnaire was validated and subjected to reliability analysis. Moreover, the information about the translation is also missing.

The authors are encouraged to provide the English version of the questionnaire as a supplementary file.

Elaborate more about the sections of the questionnaire, how many questions were present in each section. were all the questions followed liker scale assessment method? 

The scoring system of each item in the questionnaire is not described in the method section.

Results

The questionnaire was only piloted among 23 participants, it is not clear how many of them were physicians, nurses or other healthcare staff.

There is no information on the sampling method. How the participants were approached, how they were selected, what were the inclusion and exclusion criteria etc. 

The terms effective, ineffective and ambivalent are not defined in the method section. The authors are encouraged to make a heading of operational definition in the methods section and provide the various definitions used throughout the manuscript. 

This manuscript has quite a big results section due to numerous figures. I have noticed that all the information provided in the bar chart can be presented on one table. The authors are encouraged to consider concise approach to present the results. Some results are also duplicated and presented in tables, figures and text at the same time. The authors may consider additional analysis to summarize the large data. For example, the relative importance index (RII) can be helpful to understand which method is ranked first by the respondents, then second and so on...

The further evaluation of the manuscript requires correction of the methods and results section first. Moreover, mention all the tests in the methods section that have been used throughout the analysis. 

The limited section needs more sentences i.e. selection bias, sampling bias, generalizability etc.

Please separate conclusions from the implications.

Author Response

Thank you for the invitation. This manuscript is timely and interesting but lacks rigor in the methods and presentation of the results. I have few comments below.

Thank you very much for your careful reading of our paper and for your many essential suggestions, which will improve the article.

Abstract: This section does not provide quantitative results. The authors are encouraged to provide the details of major results. Please remove the numbering before each heading. The method section also needs further elaboration.

We have added essential percentage indications to the Abstract, Lines: 21-25. We have removed the numbering before each section. We have made the Methods description as complete as the number of characters in the Abstract allowed, Lines:18-21

Title:

Is it necessary to put Garlic in the title? As I have seen that the authors have evaluated various methods in this research, but only garlic was mentioned in the title?

The juxtaposition of a scientifically validated method of preventing infection against influenza, i.e., the vaccine, with a non-scientific method of preventing influenza, was deliberate to show the scale of choice and to highlight the results of the study, indicating the popularity (especially among nurses, of the latter method).

Introduction.

The rationale of this study is not clear. Is there any study on the similar topic in Poland or any other European country? If not, please state in this section. However, the authors are required to demonstrate why this study is needed and what new information is added to the existing local and international literature.

We demonstrate the rationale for choosing the topic of the study and how important it is to know the attitudes and knowledge of HWs toward ways to prevent influenza infection in the introduction of Lines: 50-63. We have added a sentence that emphasizes that to date, no such studies have been conducted in this group of people in Poland, Lines 64-66.

Methods:

The sample size heading does not provide any information on how the sample size was estimated.

After this comment, we included an indication of sample estimation in the Statistical Analysis section but moved it to the Sample Size and Characteristics subsection. We have moved this indication to Lines 95-96.

It is not clear whether the questionnaire was validated and subjected to reliability analysis. Moreover, the information about the translation is also missing.

The questionnaire used in the study was explicitly made for this study. As such, it was not translated from another language. The authors of the questionnaire, who are also the authors of the article, have a great deal of methodological experience in survey design and implementation. Based on their experience, we subjected the questionnaire to evaluation on a selected group of respondents, as we write about in The Questionnaire. Considering that it may be unclear, we added a sentence to clarify Lines: 113-119.

The authors are encouraged to provide the English version of the questionnaire as a supplementary file.

The English version of the questionnaire was included with the review response.

Elaborate more about the sections of the questionnaire, how many questions were presented in each section. were all the questions followed liker scale assessment method? The scoring system of each item in the questionnaire is not described in the method section.

Thank you very much for this comment. We have included detailed information in Lines: 97-111.

Results

The questionnaire was only piloted among 23 participants, it is not clear how many of them were physicians, nurses or other healthcare staff.

Lines 113-114

There is no information on the sampling method. How the participants were approached, how they were selected, what were the inclusion and exclusion criteria etc.

The subjects were selected using a random-target method. Before the implementation of the study, three groups of respondents have distinguished: physicians, nurses, and other hospital employees. The survey was implemented in all hospital departments to increase the diversity of those taking part in the study. One of the study's authors implemented the survey and personally contacted and surveyed the respondents. Quotas of respondents were also considered during the survey implementation. We ensured that the percentage of respondents in each of the three groups was close to the percentage of each group in the general survey population. We added this clarification. Lines 96-103

The terms effective, ineffective and ambivalent are not defined in the method section. The authors are encouraged to make a heading of operational definition in the methods section and provide the various definitions used throughout the manuscript.

Many thanks for this comment. The extremes of the scale used in this question refer to the performance of specific methods. Any issues that pertain to their effectiveness have been explained in the extensive discussion. Therefore, we see no need to build a special section for operational definitions, as it would duplicate the discussion.

This manuscript has quite a big results section due to numerous figures. I have noticed that all the information provided in the bar chart can be presented on one table. The authors are encouraged to consider concise approach to present the results. Some results are also duplicated and presented in tables, figures and text at the same time. The authors may consider additional analysis to summarize the large data. For example, the relative importance index (RII) can be helpful to understand which method is ranked first by the respondents, then second and so on....

Thank you very much for this comment. It is essential, especially since there are many results. While working on the manuscript, we were constrained by the presentation of the results. The form presented is optimal for the analysis presented. We put it together in such a way as to make it as readable as possible. Considering the rules of publication in this type of journal, we were cautious that the results in the text were not repeated in the tables.

The limited section needs more sentences i.e. selection bias, sampling bias, generalizability etc.

Thank you, we have added. Lines 675-678.

Please separate conclusions from the implications.

Thank you. We have included this Lines 687-692.

Round 2

Reviewer 2 Report

Thank you for clarifying the queries.